# USP41 Enhances Epithelial–Mesenchymal Transition of Breast Cancer Cells through Snail Stabilization

**DOI:** 10.3390/ijms24021693

**Published:** 2023-01-15

**Authors:** Ji-Yun Yoon, Seung-Un Seo, Seon-Min Woo, Taeg-Kyu Kwon

**Affiliations:** 1Department of Immunology, School of Medicine, Keimyung University, Daegu 42601, Republic of Korea; 2Center for Forensic Pharmaceutical Science, Keimyung University, Daegu 42601, Republic of Korea

**Keywords:** deubiquitinase, USP41, snail, EMT, breast cancer

## Abstract

Ubiquitination, one of many post-translational modifications, causes proteasome-mediated protein degradation by attaching ubiquitin to target proteins. Multiple deubiquitinases inhibit the ubiquitination pathway by removing the ubiquitin chain from protein, thus contributing to the stabilization of substrates. USP41 contributes to invasion, apoptosis and drug resistance in breast and lung cancer cells. However, the detailed mechanism and role of USP41 in breast cancer have not been elucidated. USP41 was overexpressed and showed poor prognosis according to the aggressive phenotype of breast cancer cells. Knockdown of USP41 inhibited migration and growth of breast cancer cells, whereas overexpression of USP41 increased cell growth and migration. In addition, depletion of USP41 downregulated Snail protein expression, an epithelial–mesenchymal transition marker, but not mRNA expression. Furthermore, USP41 interacted with and inhibited ubiquitination of Snail, resulting in the increase in Snail stabilization. Therefore, these data demonstrated that USP41 increases migration of breast cancer cells through Snail stabilization.

## 1. Introduction

Epithelial–mesenchymal transition (EMT) is a biological process where epithelial cells change into mesenchymal cells, thereby adapting the characteristics with no polarity and loss of cell adhesion [1]. Because the EMT process is also related to drug resistance, modulation of EMT is important for cancer therapy [1]. Diverse EMT transcription factors (EMT-TFs) not only suppress the expression of epithelial genes including Lamin1, E-cadherin and Claudin, but also activate mesenchymal genes such as Fibronectin and Vimentin [2]. EMT-TFs contain Snail family proteins, Zinc finger e-box binding (Zeb) homeobox family proteins and Twist family proteins [2]. Above all, Snail is modulated by various signal pathways at the transcriptional and post-translational levels [3,4]. Snail is degraded by the E3 ligase-dependent ubiquitin proteasome system [5]. Zhou et al. previously identified that β-TrCP1 ubiquitinates Snail through its phosphorylation by GSK-3β [6]. Multiple studies have since reported that F-box (FBX) family proteins (FBXL14, FBXO45, FBXO11, FBXO22, FBXW7, FBXL5 and FBXW7) inhibit EMT, metastasis and progression of breast cancer, gastric cancer and non-small lung cancer through proteasome-mediated Snail degradation [7,8,9,10,11,12,13]. In addition, TRIM21 is another E3 ligase regulating Snail ubiquitination resulting in the decrease of EMT in breast cancer cells [14].

This post-translational modification process is reversed by deubiquitinases (DUBs), which antagonize ubiquitin conjugation [15]. DUBs can be divided into six families: ubiquitin-specific proteases (USPs), ubiquitin carboxy-terminal hydrolases (UCHs), Machado–Josephin domain-containing proteases (MJDs), ovarian tumor proteases (OTUs), motif-interacting with ubiquitin-containing novel DUB family (MINDYs) and JAB1, MPN, MOV34 family (JAMMs). Previous reports investigated the involvement of DUBs in cancer metastasis through the regulation of Snail expression. For example, various USP DUB family proteins including USP3, USP26, USP29, USP37 and DUB3 specifically increase Snail deubiquitination and stabilization by interacting with Snail in breast cancer, lung cancer, gastric cancer and glioblastomas [16,17,18,19,20]. In addition, OTUB1 (an OTU family protein) and PSMD14 (a JAMM family protein) promotes metastasis of human esophageal squamous carcinoma by stabilizing Snail [21,22]. Thus, DUBs play a role as a crucial regulator in Snail-mediated cancer metastasis.

Recent studies have analyzed overall survival of cancer patients according to expression of USP41 based on The Cancer Genome Atlas (TCGA) database and suggested USP41 as marker predicting prognosis especially in invasive breast cancer and lung cancer [23,24]. Even though some researchers have reported that USP41 promotes cancer cell migration and proliferation [25,26], the underlying specific mechanism to cancer migration and EMT is still unclear.

In this study, we proposed that USP41 may be a novel DUB targeting Snail that regulates EMT and migration of breast cancer cells.

## 2. Results

### 2.1. USP41 Is Highly Expressed in Breast Tumor Tissues

Breast cancer cell lines can be classified into four subtypes; luminal A, luminal B, HER2^+^ and triple negative (TN) based on the status of three hormonal receptors, namely estrogen receptor (ER), progesterone receptor (PR) and human epithelial receptor 2 (HER2) [27]. Thus, we evaluated the USP41 level in breast cancer cell lines. The Chin breast cancer datasets were obtained from the Xena database. USP41 mRNA expression was higher in luminal B, HER2^+^ and TN of breast cancer cells compared to normal breast cells (Figure 1A). Most of all, HER2^+^ and TN breast cancer cells showed poor prognosis with the disease specific survival according to USP41 expression and aggressive phenotype (Figure 1B). Moreover, we analyzed the USP41 expression and overall survival rate of breast carcinoma using the TCGA dataset. Tumor tissues of breast cancer patients were overexpressed USP41 expression and had shorter survival time than patients with low expression of USP41 (Figure 1C,D). Thus, these data supported the potential role of USP41 in aggressive phenotype of breast cancer and bad survival rate of breast patients.

### 2.2. Knockdown of USP41 Inhibits Migration and Proliferation of Breast Cancer Cells

To investigate the role of USP41 in the migration of breast cancer cells, we examined knockdown of USP41 by siRNA in both cell lines of TNBC (MDA-MB-231) and HER^+^ (MDA-MB-361) (Figure 2A). Knockdown of USP41 suppressed wound healing activity and migration in two breast cancer cells (Figure 2B,C). Moreover, we performed colony formation and cell counting for proliferation. siRNA-mediated USP41 downregulation led to decreased colony forming ability and cell growth (Figure 2D). These data suggested that USP41 is an important role in the migration and proliferation of breast cancer cell lines.

### 2.3. Overexpression of USP41 Enhances Migration and Proliferation of Breast Cancer Cells

Next, we examined the effect of USP41 overexpression on migration and proliferation in breast cancer cells. As expected, USP41 transfected cells showed high levels of USP41 protein expression (Figure 3A). Ectopic expression of USP41 promoted cell migration and invasion in MDA-MB-231 and MDA-MB-361 cells (Figure 3B,C). Moreover, cell growth of breast cancer cells was improved by overexpression of USP41 (Figure 3D). Together, these data suggest that USP41 overexpression could enhance the growth and invasion of breast cancer cells.

### 2.4. Depletion of USP41 Destabilizes Snail

To identify the alteration of EMT-related proteins by USP41, we explored the expression of EMT-related proteins using siRNA and plasmid of USP41. Downregulation of USP41 only decreased the protein level of Snail (Figure 4A,B), as a transcription repressor in the EMT process, whereas other EMT-related proteins (E-cadherin, Vimentin and Slug) were not altered (Figure 4A). In addition, ectopic expression of USP41 upregulated Snail protein expression (Figure 4C). However, Snail mRNA expression was not altered by knockdown of USP41 (Figure 4D).

Next, we investigated the USP41-mediated Snail protein stability using cycloheximide (CHX; protein synthesis inhibitor). Treatment with CHX in USP41-depleted cells greater accelerated the degradation of Snail protein from 1 h (Figure 4E). To confirm the post-translation regulation of Snail by USP41, we examined the involvement of the ubiquitin–proteasome system using a specific proteasome inhibitor (MG132). As shown in Figure 4F, MG132 inhibited USP41 siRNA-induced Snail downregulation in MDA-MB-231 and MDA-MB-361 cells. To prove the stabilization of Snail through the enzymatic activity of USP41, we constructed the catalytic inactive USP41 mutant (C42A). USP41 wild-type (WT) sustained Snail protein expression, whereas USP41 mutant (C42A) degraded more compared to that of vector-transfected cells (Figure 4G). These data demonstrated that USP41 induces proteasome-mediated Snail protein stabilization.

### 2.5. USP41 Interacts with and Deubiquitinates Snail

Since DUBs regulate various cellular processes by interacting with target proteins [28], we examined the interaction between USP41 and Snail. When we performed immunoprecipitation (IP) assay using Snail antibody, we identified that Snail interacts with USP41 in both breast cancer cells (Figure 5A). Moreover, siRNA-mediated USP41 downregulation increased the ubiquitination of Snail in both breast cancer cells (Figure 5B), whereas overexpression of USP41 decreased Snail ubiquitination (Figure 5C). Therefore, these results indicated that USP41 interacts with Snail resulting in the induction of Snail destabilization.

Because TRIM21 is a key E3 ligase in the ubiquitination and degradation of Snail [14], we investigated the involvement of TRIM21 in USP41 depletion-mediated Snail degradation. Indeed, siRNA of TRIM21 alone increased Snail protein level (Figure 5D). Knockdown of TRIM21 markedly rescued Snail downregulation by USP41 siRNA in both breast cancer cells. Moreover, we found the interaction of USP41 and TRIM21 through Snail IP assay using USP41 antibody (Figure 5E). Collectively, these results suggest that Snail ubiquitination is modulated by the expression levels of USP41 and TRIM21.

### 2.6. USP41 Regulates Migration and Proliferation of Breast Cancer Cells through Snail

To verify the involvement of Snail in USP41-mediated EMT regulation, we executed the overexpression of Snail in USP41-depleted cells. Undoubtedly, Snail overexpression alone increased migration of breast cancer cells (Figure 6A–C). Upregulation of Snail improved the decrease in cancer cell migration by USP41 knockdown (Figure 6B,C). In contrast, depletion of Snail suppressed cell migration in USP41-overexpressed breast cancer cells (Figure 6D–F). Moreover, we examined the role of Snail in the proliferation of breast cancer cells according to USP41 expression. Likewise, the ectopic expression of Snail prevented the USP41 siRNA-mediated reduction of cancer cell proliferation, while knockdown of Snail inhibits proliferation by USP41 overexpression (Figure 6G,H). Therefore, these results suggest that Snail contributes to USP41-mediated migration and proliferation in breast cancer cells. In addition, we investigated the correlation between USP41 and Snail expression in the TCGA data base, and expression between USP41 and Snail represented a positive correlation (Figure 6I).

## 3. Discussion

In this study, we suggested the role of USP41 as a new DUB of Snail stabilization. USP41 interacts with Snail and stabilizes Snail via deubiquitination. In addition, knockdown of USP41 suppressed migration and growth of breast cancer cells. Therefore, USP41 plays a critical role in Snail-mediated EMT progression in breast cancer cells (Figure 7).

Previously, Huang et al. identified the RACK1 as a novel protein interacting with USP41 through CoIP-MS analysis and indicated the promotion of migration via USP41-induced RACK1 upregulation [25]. Since function of RACK1 is diverse and extensive in breast cancer, they mentioned that the investigation of the roles of USP41-mediated RACK1 in cancer cell migration are necessary. In addition, they did not examine the EMT-regulated proteins by USP41 knockdown or overexpression [25]. In our study, knockdown or overexpression of USP41 induced the downregulation or upregulation of Snail protein expression in breast cancer cells, respectively (Figure 4A–C).

Various targets involved in EMT are regulated by diverse E3 ligases through ubiquitination-mediated protein degradation, thereby modulating the EMT process [29]. Snail, a major transcription factor of EMT, can especially be controlled through the transcriptional or post-translational levels. Recently, TRIM21 was identified as an E3 ligase for Snail and overexpression of TRIM21 led to the degradation of Snail through ubiquitination [14]. To prove the relationship with USP41 and TRIM21 in the downregulation of Snail, we performed single or double knockdown of these genes. USP41 knockdown-mediated Snail downregulation was prevented by TRIM21 knockdown (Figure 5D). Therefore, we supposed that the balance of USP41 and TRIM21 may modulate the ubiquitination and stabilization of Snail. 

Snail acts as a transcription factor of the EMT-regulated gene, thereby modulating tumor metastasis [30]. Snail represses E-cadherin, an epithelial marker, by binding to the promoter of E-cadherin, whereas mesenchymal markers, such as Vimentin, Slug and Fibronectin, are increased by Snail [31,32]. Previously, Ji et al. reported that shRNA-mediated USP41 knockout suppresses migration via E-cadherin upregulation in lung adenocarcinoma [26]. However, our data did not show the alteration of E-cadherin expression by USP1 siRNA (Figure 4A). This discrepancy is thought to be due to differences in lines and/or cell contexture. In addition, mesenchymal-related proteins (Vimentin and Slug) were not changed in USP41-depleted cells despite the downregulation of Snail (Figure 4A). Therefore, more work is needed to understand which genes whose transcription is regulated by Snail are altered following USP41 knockdown.

Recently, Li et al. suggested that USP41 is one of prediction genes of invasive breast cancer prognosis through bioinformatic analysis [24]. They emphasize the positive correlation of USP41 and prognosis in patients with invasive breast cancer. We also found that USP41 is highly expressed and shows poor prognosis depending on invasiveness of breast cancer cell lines (Figure 1A,B). Therefore, USP41 can act as a tumor biomarker, thereby influencing prognostic prediction and clinical treatment of breast cancer patients.

Collectively, this study showed that USP41 inhibition suppresses the migration of breast cancer cells through Snail downregulation. Therefore, USP41 could be used as a crucial target for cancer therapy, given its role in the inhibition of breast cancer metastasis.

## 4. Materials and Methods

### 4.1. Cell Lines and Cell Culture

Human breast carcinoma cells (MDA-MB-231, MDA-MB-361) were purchased from American Type Culture Collection (Manassas, VA, USA). The cells were maintained in Dulbecco’s Modified Eagle Media (DMEM) supplemented with 10% fetal bovine serum (Welgene, Gyeongsan, Republic of Korea), 1% antibiotic-antimycotic (Gibco, Waltham, MA, USA) and 0.2% gentamicin (Thermo Fisher Scientific, Waltham, MA, USA). The cells were cultured at 37 °C, 5% CO_2_ in humidified air.

### 4.2. Antibodies and Reagents

Thermo Fisher Scientific supplied the anti-USP41 antibody (Waltham, MA, USA). Anti-Snail, anti-E-cadherin, anti-Vimentin and anti-Slug antibodies purchased from Cell signaling (Beverly, MA, USA) were used for Western blotting. Anti-Actin antibody, cycloheximide and MG132 supplied by Sigma Chemical Co. (St. Louis, MO, USA) were used for the experiment. Enzo Life Sciences supplied anti-Ub antibody (San Diego, CA, USA). Santa Cruz Biotechnology supplied protein G agarose bead (St. Louis, MO, USA). TriZol was purchased from Life Technologies (Gaithersburg, MD, USA). Immobilon Western chemiluminescent HRP substrate was supplied by Merck Millipore (Burlington, MA, USA).

### 4.3. Analysis of USP41 Expression and Survival Rate

To analyze the USP41 expression and survival rate in breast cancer cell type and normal and tumor cells with breast patients, we obtained TCGA database from Xena (https://xena.ucsc.edu/ accessed on 20 November 2022).

### 4.4. Transfection

Control siRNA and USP41 siRNA (#1; 5′-GGA AGA AGA CCC GUG GGA A-3′ and #2; 5′-CCU GCU GCC UUA ACU CCU U-3′) obtained from Bioneer (Daejeon, Republic of Korea) were used with Lipofectamine^®^ RNAiMAX Reagent (Invirtrogen, Calshad, CA, USA) for siRNA transfection. pcDNA 3.1 vector was obtained from Invitrogen (Carlsbad, CA, USA). pCMV-USP41-FLAG plasmid was obtained from the life sciences market (Dundas St, Mong Kok, Hong Kong, China). For plasmid transfection, Lipidofect-P was used (Lipidomia, Seongnam, Republic of Korea).

### 4.5. Western Blotting

Cells were dissolved in RIPA lysis buffer supplemented with 20 mM HEPES and 0.5% Triton X-100, pH 7.6. Proteins were separated and transferred by electrophoresis to the nitrocellulose membranes (GE Healthcare life Science, Pittsburgh, PO, USA). The membranes were blocked with 5% non-fat milk containing 0.02% sodium azide, and then reacted with a primary and secondary antibody step by step. After washing with 1 X tris buffered saline with tween, images of protein band were captured by immobilon Western chemiluminescent HRP substrate and iBrightTM Imager (Thermo Fisher Scientific, Waltham, MA, USA).

### 4.6. Cancer Cell Migration

Migration ability was examined by wound healing assay; after seeding in a 12-well plate for 24 h, the cells were scratched by a yellow tip and the medium was changed to DMEM containing 1% FBS. Cells of 1 × 10^5^ were seeded in the upper part of transwell with DMEM and the bottom of the transwell with DMEM containing 1% FBS. After 24 h incubation, cells were fixed with methanol at −20 °C for 10 min and washed with distilled water twice. The top surface of the cells of the transwell were removed using a cotton swab. Cells that transferred to under the surface of the transwell were stained with 0.25% crystal violet for 20 min and washed with distilled water.

### 4.7. Colony Formation

After seeding in a 6-well plate, cells were incubated for 3 days. Then, after washing with phosphate buffered saline (PBS), cells were stained with 0.25% crystal violet for 5 min at room temperature. After washing them with PBS, the buffer was removed.

### 4.8. Quantitative PCR (qPCR)

cDNA was produced using RNA extracted by TriZol and M-MLV reverse transcriptase (Gibco-BRL, Gaithersburg, MD, USA). After dilution of cDNA, it reacted with SYBR Fast qPCR Mix (Takara Bio Inc., Shiga, Japan) and primers. For qPCR analysis, Thermal Cycler Dice^®^ Real-Time System III (Takara Bio Inc., Shiga, Japan) was used. The primers were used as follows: Actin (forward) 5′-CTA CAA TGA GCT GCG TGT G-3′ and (reverse) 5′-TGG GGT GTT GAA GGT CTC-3′; and Snail (forward) 5′-TTT CTG GTT CTG TGT CCT CTG CCT-3′ and (reverse) 5′-TGA GTC TGT CAG CCT TTG TCC TGT-3′.

### 4.9. Immunoprecipitation (IP) Assay

After the harvest, cells were washed with PBS and dissolved using RIPA buffer. After sonication, the cells were centrifuged and the supernatants reacted with primary antibody overnight. After adding protein G agarose beads to the lysates, it reacted. After centrifuging, the supernatants were removed. Lysates were washed with RIPA lysis buffer, and then boiled adding the 2 × cooking buffer. The interaction between the protein was identified by Western blotting.

### 4.10. Ubiquitination Assay

Cells were co-transfected with HA-UB plasmid and control siRNA or USP41 siRNA using Lipofectamine^TM 2000^ (Invitrogen, Carlsbad, CA, USA). The transfected cells were treated with MG132. After harvest, cells were washed with PBS containing 10 mM N-ethylmaleimide (NEM), resuspended in PBS containing 10 mM NEM. 1% SDS was added to the lysates and boiled at 95 °C for 10 min. RIPA lysis buffer involving 1 mM PMSF, 1 mM Na_3_VO_4_, 0.01 μg/mL leupeptin, 2 mg/mL aprotinin and 5 mM NEM were added to lysates and the lysates were dissolved using syringe and centrifuged 13,000 RPM for 15 min at 4 °C. The supernatants reacted with the primary antibody overnight and added protein G agarose bead for 2 h. After centrifuging, the lysates were washed with RIPA lysis buffer and boiled after adding 2 × cooking buffer for 10 min. After centrifuging for 10 min, Western blotting was performed. Ubiquitination was identified in denaturation condition with anti-Ub antibody.

### 4.11. Statistical Analysis

The data were analyzed using a one-way ANOVA and post-hoc comparisons (Student–Newman–Keuls) using the Statistical Package for Social Sciences 22.0 software (SPSS Inc.; Chicago, IL, USA).

## 5. Conclusions

Our study suggested that USP41 knockdown inhibits proliferation and invasive ability of breast cancer cells. In mechanisms, USP41 bound to Snail, a master regulator of EMT, and induced Snail deubiquitination, thereby increasing the stabilization of Snail. USP41-mediated Snail stabilization plays a critical role in EMT and metastasis, and provides a crucial target against breast cancer therapy.

## Figures and Tables

**Figure 1 ijms-24-01693-f001:**
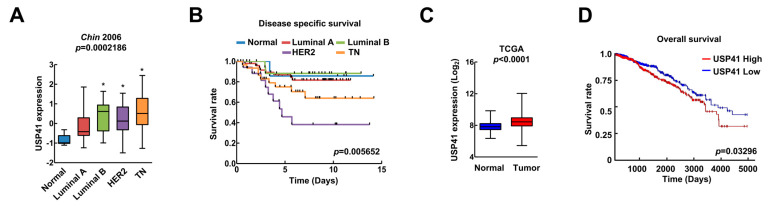
High expression of USP41 has poor prognosis in breast cancer patients. (**A**,**B**) Analysis of USP41 mRNA expression (**A**) and survival rate (**B**) according to hormonal receptor and HER2/neu status using the breast cancer dataset of Chin et al. from Xena (https://xena.ucsc.edu/ accessed on 20 November 2022). (**C**,**D**) Analysis of USP41 mRNA expression (**C**) and survival rate (**D**) using TCGA dataset in breast carcinoma. * *p* = 0.0002186 compared to the Normal.

**Figure 2 ijms-24-01693-f002:**
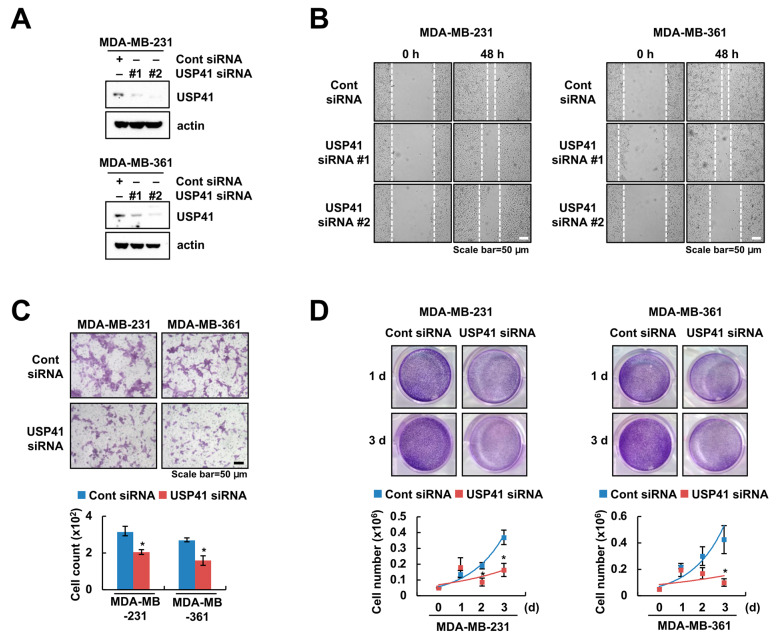
Knockdown of USP41 inhibits migration and proliferation of breast cancer cells. (**A**–**D**) MDA-MB-231 and MDA-MB-361 cells were transfected using control (Cont) and USP41 siRNA for 48 h. The protein levels of USP41 were examined by Western blotting (**A**). Migration ability was examined by wound healing assay (**B**) and transwell assay (**C**). Colony formation and cell growth were determined using crystal violet staining and cell counting, respectively (**D**). The values in the graph represent the mean ± SD of three independent samples, respectively. * *p* < 0.05 compared to the control siRNA.

**Figure 3 ijms-24-01693-f003:**
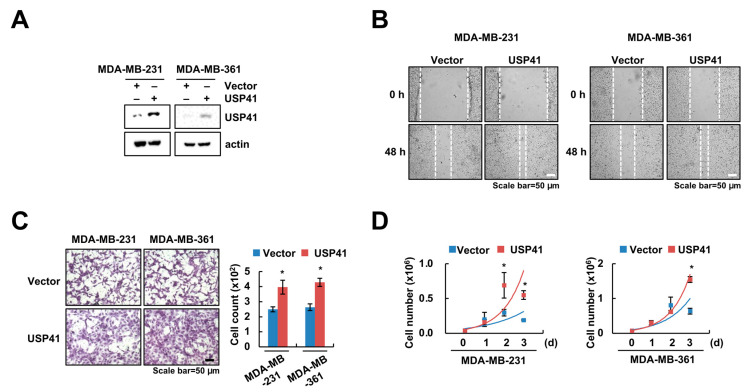
Overexpression of USP41 increases migration and proliferation of breast cancer cells. (**A**–**D**) MDA-MB-231 and MDA-MB-361 cells were transfected using vector and USP41 plasmid for 48 h. The protein levels of USP41 were examined by Western blotting (**A**). Migration ability was examined by wound healing assay (**B**) and transwell assay (**C**). Cell growth was determined by cell counting (**D**). The values in the graph represent the mean ± SD of three independent samples, respectively. * *p* < 0.05 compared to the vector.

**Figure 4 ijms-24-01693-f004:**
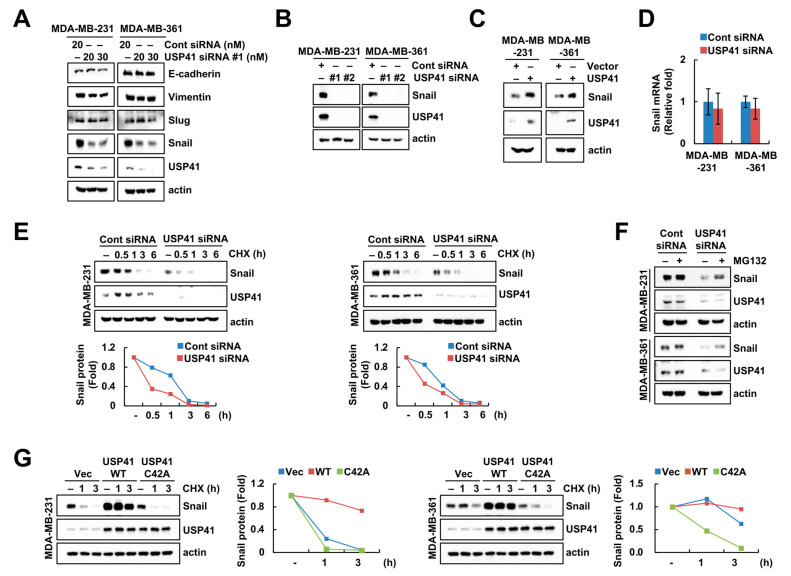
Depletion of USP41 downregulates Snail expression at the post-translational level. (**A**–**D**) MDA-MB-231 and MDA-MB-361 cells were transfected using Cont siRNA and USP41 siRNA (**A**,**B**,**D**), or vector and USP41 plasmid (**C**) for 48 h. (**E**) MDA-MB-231 and MDA-MB-361 cells were transfected with Cont or USP41 siRNA, and treated with 20 μg/mL cycloheximide (CHX) for the indicated time points. (**F**) MDA-MB-231 and MDA-MB-361 cells were transfected with Cont or USP41 siRNA, and treated with 0.25 μM MG132 for 12 h. (**G**) MDA-MB-231 and MDA-MB-361 cells were transfected with vector, USP41 wild-type (WT) or mutant (C42A), and treated with 20 μg/mL CHX for the indicated time points. Protein (**A**–**C**,**E**–**G**) and mRNA expression (**D**) were determined by Western blotting and qPCR, respectively. The density of Snail protein was quantified by Image J program.

**Figure 5 ijms-24-01693-f005:**
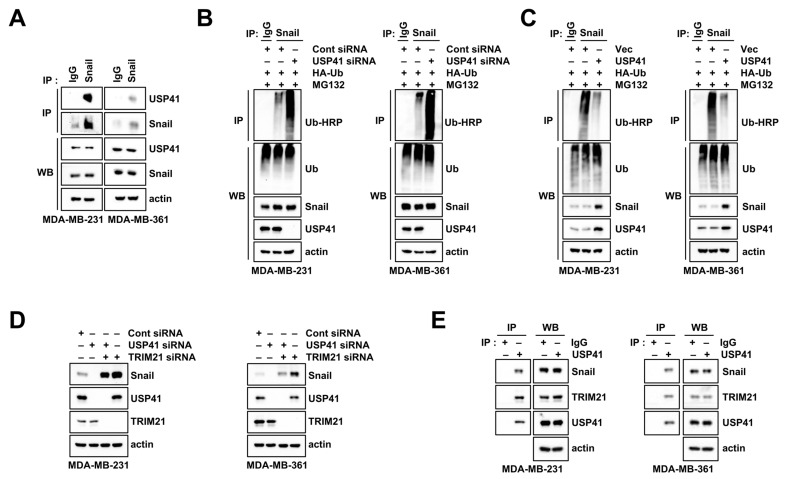
USP41 interacts and deubiquitylates with Snail. (**A**) Interaction between endogenous USP41 and Snail was examined by immunoprecipitation (IP) assay. (**B**,**C**) MDA-MB-231 and MDA-MB-361 cells were co-transfected with Cont and USP41 siRNA (**B**) or vector and USP41 (**C**) with HA-Ub plasmid, and treated with 0.25 μM MG132 for 24 h. (**D**) MDA-MB-231 and MDA-MB-361 cells were transfected with Cont, USP41, or/and TRIM21 siRNA for 48 h. (**E**) Interaction between endogenous Snail or TRIM21 with USP41 was examined by IP assay. IP (**A**,**E**) and ubiquitination (**B**,**C**) were detected by Western blotting. Protein expression was determined by Western blotting (**D**).

**Figure 6 ijms-24-01693-f006:**
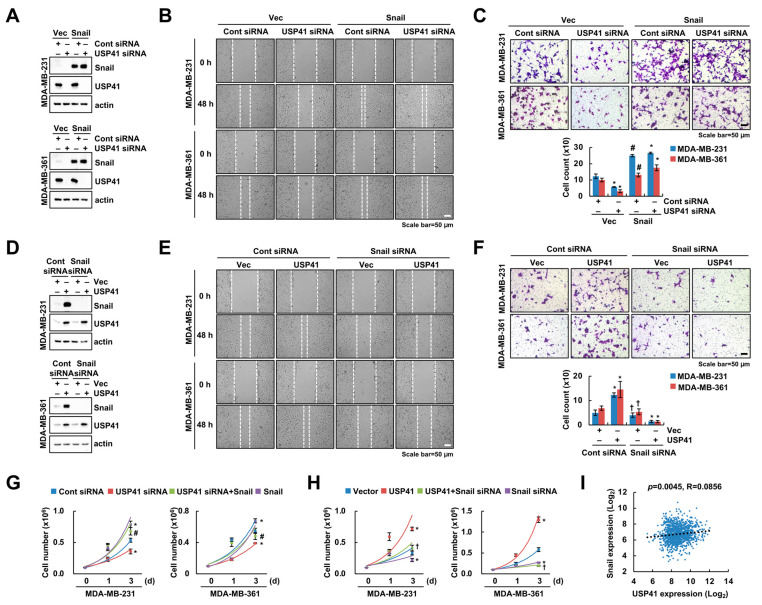
USP41-dependent Snail regulation is involved in breast cancer migration. (**A**–**C**,**G**) MDA-MB-231 and MDA-MB-361 cells were co-transfected with USP41 siRNA and Snail plasmid for 48 h. (**D**–**F**,**H**) MDA-MB-231 and MDA-MB-361 cells were co-transfected with USP41 plasmid and Snail siRNA for 48 h. The protein levels were examined by Western blotting (**A**,**D**). Migration ability was examined by wound healing assay (**B**,**E**) and transwell assay (**C**,**F**). Cell growth was determined by cell counting (**G**,**H**). (**I**) Analysis of correlation with USP41 and Snail mRNA expression using TCGA dataset in breast carcinoma. The values in the graph represent the mean ± SD of three independent samples, respectively. ** p* < 0.05 compared to the vector plus Cont siRNA-transfected cells. ^#^ *p* < 0.05 compared to the USP41 siRNA. ^†^ *p* < 0.05 compared to the USP41 overexpression.

**Figure 7 ijms-24-01693-f007:**
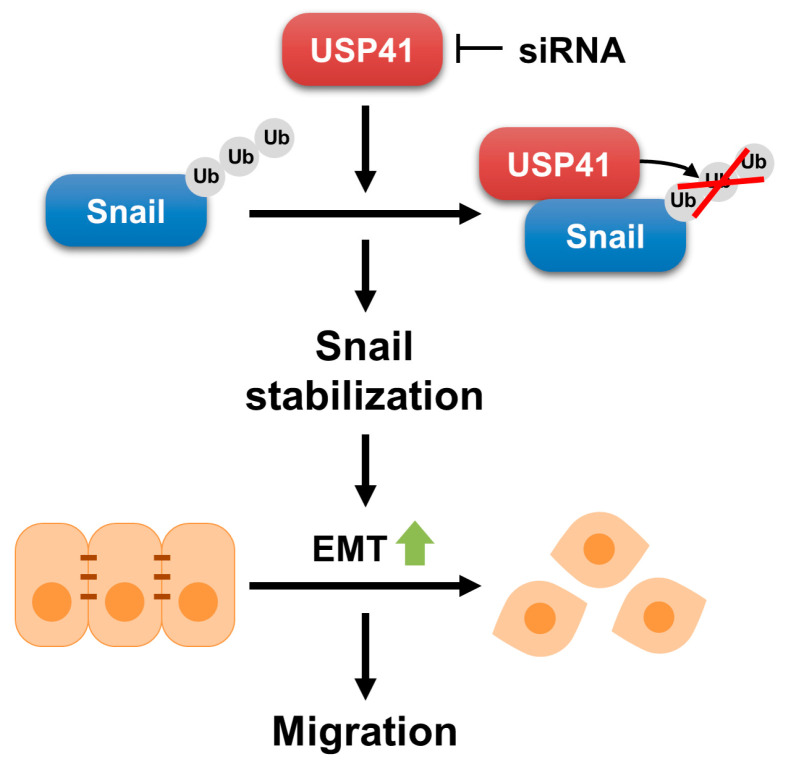
Schematic illustration showing the role and mechanism of USP41 in Snail stabilization and EMT.

## Data Availability

Not applicable.

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
