# Peer review of "USP41 Enhances Epithelial–Mesenchymal Transition of Breast Cancer Cells through Snail Stabilization"

_ijms, 2023, doi:10.3390/ijms24021693_

Round 1

Reviewer 1 Report

In this study, Ji Yun Yoon et al. identified that USP41 is a novel deubiquitinase of Snail and promotes migration of breast cancer cells. USP41 has been reported to enhance breast cancer proliferation and migration via regulating RACK1. Additionally, TRIM21 was identified to be a Dub of Snail in breast cancer cells, mediating ubiquitination of Snail and modulates EMT. Therefore, the novelty of this study is very limited. However, it is possible that Snail is regulated by various Dub at different physiological and pathological condition. Identifying individual Dub of Snail can help understand the underlying mechanism of Snail-mediated cancer metastasis. An important question of this study is the relationship of the function of USP41 and TRIM21 on regulating Snail in breast cancer. Do they have Synergistic or antagonistic effects? Besides, I have some other detailed comments as follows to improve this study.

1.     The conclusion of this study is that USP41 increases migration of breast cancer cells through Snail stabilization. However, the data only demonstrated that USP41 promotes migration and proliferation of breast cancer cells, but they did not reveal the role of snail in the process. As USP41 has many substrates, the conclusion of this study is not solid and convincing, which must be strengthened by experiments and efficient data.

2.     The data did not show the effects of overexpression of USP41 on Snail ubiquitination, which should be supplemented.

3.     It will be beneficial and meaningful to detect the protein level of USP41 and Snail, and analyze the correlation of these two proteins in human breast cancer tissues.

4.     I am confused with the text of figure 5A. Both USP41 and snail are labelled with “IP”.

5.     The Discussion is too simple and lack of depth. For example, an Inescapable question is that the possible relationship between USP41 and TRIM21 on regulating Snail in breast cancer.

Author Response

12, Jan 2023

Dear, Editor

We sincerely appreciate the time and effort of you and the referees spent in considering and evaluating our manuscript (ijms-2079542) entitled with “USP41 Enhances Epithelial-Mesenchymal Transition of Breast Cancer cells Through Snail stabilization” by Ji Yun Yoon et al., for publication in International Journal of Molecular Sciences. Having received the kind comments by reviewer, we revised our manuscript with attention to each of the comments by reviewer. We appreciate the reviewer very much, who raised the very important critiques to strengthen the claim of our manuscript. We have given very careful consideration to the suggestions and have revised our manuscript. We performed additional experiments and new information are incorporated in the revised version of our manuscript. We have responded all the comments by the referee point-by-point as follows;

  1. Respond to Reviewer #1’s comment

1) An important question of this study is the relationship of the function of USP41 and TRIM21 on regulating Snail in breast cancer. Do they have Synergistic or antagonistic effects?

Answer: Thank you for your comments. As your comments, to examine the relationship of USP41 and TRIM21 on regulation of Snail, we used single or double knockdown system of USP41 and TRIM21. We provided data in Figure 5D, E and described in Results Parts (line 158).

2) The conclusion of this study is that USP41 increases migration of breast cancer cells through Snail stabilization. However, the data only demonstrated that USP41 promotes migration and proliferation of breast cancer cells, but they did not reveal the role of snail in the process. As USP41 has many substrates, the conclusion of this study is not solid and convincing, which must be strengthened by experiments and efficient data.

Answer: Thank you for your comments. As your comments, we investigated the role of Snail in USP41-mediated migration and proliferation of breast cancer cells. we provided the data in Figure 6 and described in Results Parts (line 175).

3) The data did not show the effects of overexpression of USP41 on Snail ubiquitination, which should be supplemented.

Answer: Thank you for your comments. As your comments, we performed the ubiquitination assay of Snail by USP41 overexpression and provided the data in Figure 5C.

4) It will be beneficial and meaningful to detect the protein level of USP41 and Snail, and analyze the correlation of these two proteins in human breast cancer tissues.

Answer: Thank you for your comments. Unfortunately, we did not acquire the human breast cancer tissues. Therefore, we analyzed the correlation of USP41 and Snail in the TCGA data base instead of breast cancer tissues, and provided the data in Figure 6I.

5) I am confused with the text of figure 5A. Both USP41 and snail are labelled with “IP”.

Answer: Thank you for your comments. As your comments, we modified the label to IP experiment in Figure 5A.

6) The Discussion is too simple and lack of depth. For example, an Inescapable question is that the possible relationship between USP41 and TRIM21 on regulating Snail in breast cancer.

Answer: Thank you for your comments. As your comments, we described the relationship between USP41 and TRIM21 in regulation of Snail in Discussion Parts (line 215).

Thank you very much for your detailed and thoughtful comments. All authors concur with the submission of this revision. Please, don’t hesitate to contact with me.

Sincerely yours,

Reviewer 2 Report

In this article, the authors investigated the role of deubiquitinating enzymes (DUBs) in controlling the Epithelial-Mesenchymal Transition (EMT) through regulation of the EMT transcription factor Snail in a couple of breast cancer cell lines. EMT is characterized by the loss of endothelial cell markers and the concomitant gain of mesenchymal markers, many of them are controlled by EMT transcription factors such as Snail. Since cells undergoing EMT gain the ability to migrate and invade other tissues, understanding the molecular events controlling this process has obvious implications for cancer biology. Here, the authors focused on the regulation of EMT by the ubiquitin-proteasome system (UPS). The UPS controls protein turnover by conjugating ubiquitin chains to target proteins, directing them for degradation by the proteasome. The UPS uses an elaborate 3 step enzymatic cascade to trigger ubiquitin conjugation (ubiquitination). Importantly, protein ubiquitination can be reversed by the action of proteases called DUBs, 100 of which are encoded in the human genome and classified in 6 distinct families, with the ubiquitin specific protease family (USP) being the largest one and comprising ~50 members.

Here, Yoon, Woo & al. focused on USP41, a poorly characterized member of the USP family, to show that it is involved in cell migration as well as proliferation and propose that it acts as a DUB for the transcription factor Snail. Very little is known about USP41 biological functions, and the few published papers point to a pro-proliferative function of this DUB. Thus, this report aims at shedding some light on a molecular mechanism regulated by this uncharacterized DUB. While most of the data presented is clean, the paper in its current form feels very preliminary, and even more with every section being extremely short. Thus, it is not suitable for publication in its current form.

-          The main findings of this article are that (i) knock down or overexpression of USP41 impact cell migration and (ii) that USP41 interacts and stabilizes Snail. Because Snail is involved in cell migration, the authors conclude that USP41 controls cell migration through regulation of Snail. However, this is never clearly demonstrated. For instance, the authors should perform rescue experiments like the ones presented in figures 2 and 3, where they would overexpress Snail in cells knocked down for USP41 or knock down Snail in cells overexpressing USP41. This will show a direct link between USP41, Snail and migration, as proposed in their model in figure 6. Related to figures 2 and 3, they could both be easily merged, since they show the same type of experiments with both loss of function and gain of function approaches, which nicely complement each other.

-          Related to the knock down and overexpression experiments, they are however missing a couple of important controls. First, the authors should use a second siRNA reagent to make sure that what they are observing is not due to off target effects. Second, they should use a catalytically inactive version of USP41, where the active cysteine is mutated to serine, to show that the effects of USP41 overexpression are due to the intrinsic enzymatic activity of the DUB. Related to gain of function experiments, the authors could also consider doing the CHX experiments after USP41 overexpression, with both wild type and catalytically inactive constructs.

-          The experiment showing Snail ubiquitination has to be redone and probably optimized. While they include the right control, the fact that there is so much background in the IgG lane makes it hard to interpret. Moreover, there is always the possibility that the signal observed in these experiments show indirect ubiquitination, like ubiquitination of a Snail interacting proteins. I would suggest the authors do the opposite (IP Ub using the HA tag) and blot for Snail, where the expectation would be to see slowly migrating form of Snail indicating ubiquitin conjugation. Ideally, using a His6-tagged ubiquitin construct would be even better, since it will allow purification of ubiquitinated species under strong denaturing conditions (such as 6M Guanidine or 8M Urea) and prevent enrichment of proteins interacting nonspecifically with Ub (including potentially Snail itself).

-     -          My last comment will be about the writing. First, it is not clearly explained in the introduction why the authors focused on USP41 to begin with. They do explain it in the results, but it would be more useful if that was stated directly in the introduction. Second, and I realize that the authors are not native English speakers, there are quite a few typos all along the manuscript, and some sentences are sometimes hard to understand at first. I am listing a few examples below but would suggest the authors to have their manuscript carefully read and edited before another submission.

Line 11 “one of MANY post-translational modificationS”

Lane 13 Instead of “inducing of protein stability” maybe use something like “thus contributing to stabilization of substrates

Line 26 “where epithelial cells”

Line 27 “which” should be “with” I believe

Line 40 mention quickly that TRIM21 is another E3 regulating snail ubiquitination

Line 42 DUBs antagonize ubiquitin conjugation

Line 49 and 50 should be part of the same paragraph

Line 55 sentence should be rewritten

Line 56 Because USP41 is almost uncharacterized the authors should emphasize it more. It is really not clear why they focused on this one to begin with though.

Line 115 “did not alter” should be “were not altered”

Line 118 it’s written USP42 instead of USP41

Line 119 the sentence should be rewritten completely

Line 136 “by interaction” should be “by interacting”

Line 141 “resulting” instead of “resulted”

Line 156 “reporting that” instead of "reported the”

Line 171 Sentence should be rewritten, I believe the authors meant that more work is needed to understand which genes whose transcription is regulated by Snail are altered following USP41 knockdown.

Author Response

12, Jan 2023

Dear, Editor

We sincerely appreciate the time and effort of you and the referees spent in considering and evaluating our manuscript (ijms-2079542) entitled with “USP41 Enhances Epithelial-Mesenchymal Transition of Breast Cancer cells Through Snail stabilization” by Ji Yun Yoon et al., for publication in International Journal of Molecular Sciences. Having received the kind comments by reviewer, we revised our manuscript with attention to each of the comments by reviewer. We appreciate the reviewer very much, who raised the very important critiques to strengthen the claim of our manuscript. We have given very careful consideration to the suggestions and have revised our manuscript. We performed additional experiments and new information are incorporated in the revised version of our manuscript. We have responded all the comments by the referee point-by-point as follows;

  1. Respond to Reviewer #2s comment

1) the authors conclude that USP41 controls cell migration through regulation of Snail. However, this is never clearly demonstrated. For instance, the authors should perform rescue experiments like the ones presented in figures 2 and 3, where they would overexpress Snail in cells knocked down for USP41 or knock down Snail in cells overexpressing USP41.
Answer: Thank you for your comments. As your comments, to verify the role of Snail in USP41-mediated migration of breast cancer cells, we performed rescue experiments with loss or gain of function to Snail in overexpression or knockdown of USP41. We provided these data in Figure 6 and described in Results Parts (line 175).

2) Related to the knock down and overexpression experiments, they are however missing a couple of important controls. First, the authors should use a second siRNA reagent to make sure that what they are observing is not due to off target effects.

Answer: Thank you for your comments. As your comments, we examined migration and Snail expression using 2 type siRNA to rule out off-target effects, and provided the data in Figure 2A, B and Figure 4B.

3) Second, they should use a catalytically inactive version of USP41, where the active cysteine is mutated to serine, to show that the effects of USP41 overexpression are due to the intrinsic enzymatic activity of the DUB. Related to gain of function experiments, the authors could also consider doing the CHX experiments after USP41 overexpression, with both wild type and catalytically inactive constructs.

Answer: Thank you for your comments. As your comments, we constructed the plasmid of catalytic inactive USP41 mutant (C42A) and performed the Snail protein stability experiment using CHX. As expected, wild type of USP41 sustained the Snail expression, whereas USP41 C42A more degraded Snail expression. We provided the data in Figure 5C and described in Results Part (line 133).

4) The experiment showing Snail ubiquitination has to be redone and probably optimized. While they include the right control, the fact that there is so much background in the IgG lane makes it hard to interpret. Ideally, using a His6-tagged ubiquitin construct would be even better, since it will allow purification of ubiquitinated species under strong denaturing conditions

Answer: Thank you for your comments. We performed the ubiquitination assay under denaturation conditions. As your comments, we re-investigated Snail ubiquitination by USP41 knockdown under denaturation conditions, and provided the data in Figure 5B.

5) First, it is not clearly explained in the introduction why the authors focused on USP41 to begin with. They do explain it in the results, but it would be more useful if that was stated directly in the introduction.

Answer: Thank you for your comments. As your comments, we explained the reason of focused on USP41 in Introduction Part (line 57).

6) Second, and I realize that the authors are not native English speakers, there are quite a few typos all along the manuscript. I am listing a few examples below but would suggest the authors to have their manuscript carefully read and edited before another submission.

Answer: Thank you for your comments. As your comments, we edited manuscripts.

6-1) Line 11 “one of MANY post-translational modificationS”

Answer: Thank you for your comments. As your comments, we added ‘many’ (line 12).

6-2) Lane 13 Instead of “inducing of protein stability” maybe use something like “thus contributing to stabilization of substrates

Answer: Thank you for your comments. As your comments, we modified (line 14).

6-3) Line 26 “where epithelial cells”

Answer: Thank you for your comments. As your comments, we modified (line 27).

6-4) Line 27 “which” should be “with” I believe

Answer: Thank you for your comments. As your comments, we modified (line 28).

6-5) Line 40 mention quickly that TRIM21 is another E3 regulating snail ubiquitination

Answer: Thank you for your comments. As your comments, we modified (line 41).

6-6) Line 42 DUBs antagonize ubiquitin conjugation

Answer: Thank you for your comments. As your comments, we modified (line 44).

6-7) Line 49 and 50 should be part of the same paragraph

Answer: Thank you for your comments. As your comments, we modified (line 50).

6-8) Line 55 sentence should be rewritten

Answer: Thank you for your comments. As your comments, we rewritten this sentence (line 55).

6-9) Line 56 Because USP41 is almost uncharacterized the authors should emphasize it more. It is really not clear why they focused on this one to begin with though.

Answer: Thank you for your comments. As your comments, we supplemented the reason of focused on USP41 in cancer migration (line 57).

6-10) Line 115 “did not alter” should be “were not altered”

Answer: Thank you for your comments. As your comments, we modified (line 124).

6-11) Line 118 it’s written USP42 instead of USP41

Answer: Thank you for your comments. As your comments, we modified (line 126).

6-12) Line 119 the sentence should be rewritten completely

Answer: Thank you for your comments. As your comments, we rewritten this sentence (line 127).

6-13) Line 136 “by interaction” should be “by interacting”

Answer: Thank you for your comments. As your comments, we modified (line 150).

6-14) Line 141 “resulting” instead of “resulted”

Answer: Thank you for your comments. As your comments, we modified (line 156).

6-15) Line 156 “reporting that” instead of "reported the”

Answer: Thank you for your comments. As your comments, we modified.

6-16) Line 171 Sentence should be rewritten, I believe the authors meant that more work is needed to understand which genes whose transcription is regulated by Snail are altered following USP41 knockdown.

Answer: Thank you for your comments. As your comments, we modified (line 224).

Thank you very much for your detailed and thoughtful comments. All authors concur with the submission of this revision. Please, don’t hesitate to contact with me.

Sincerely yours,

Round 2

Reviewer 1 Report

The authors have addressed my major concerns and greatly improved the manuscript. However, a new result puzzled me. The author investigated the correlation between USP41 and Snail expression in TCGA database and found that the mRNA expression levels of USP41 and Snail are positively correlated. The authors have demonstrated that USP41 is the DUB of Snail, so the regulation of USP41 on Snail is post-translational modification. Why is the mRNA level of USP41 positively correlated with Snail? Does it mean that USP41 also regulates Snail at the mRNA level? It is an important question to explain and discuss.

Author Response

Thank you for your comments. Unfortunately, we did not acquire the human breast cancer tissues. Therefore, we analyzed the correlation of USP41 and Snail in the TCGA data base instead of breast cancer tissues, and provided the data in Figure 6I.

In addition,  TCGA data analysis is correct to provide mRNA levels. We only analyzed the correlation between USP41 mRNA and Snail mRNA expression levels. However, the function of USP41 protein does not act as a transcription factor, but there is a possibility that the transcription factor among the USP41 target proteins is regulated to regulate Snail mRNA expression. Most importantly, it is best to investigate the correlation between USP41 and Snail protein in cancer patient samples. Sorry for not being able to provide these results unfortunately. However, our results provided sufficient information on the mechanism of Snail stabilization by USP41.